# The ER membrane protein complex is required to ensure correct topology and stable expression of flavivirus polyproteins

Ashley M Ngo[1], Matthew J Shurtleff[2], Katerina D Popova[2], Jessie Kulsuptrakul[1], Jonathan S Weissman[2,3], Andreas S Puschnik[1]*

[1]Chan Zuckerberg Biohub, San Francisco, United States; [2]Department of Cellular and Molecular Pharmacology, University of California, San Francisco, San Francisco, United States; [3]Howard Hughes Medical Institute, University of California, San Francisco, San Francisco, United States

*For correspondence:
andreas.puschnik@czbiohub.org

**Competing interests:** The authors declare that no competing interests exist.

**Abstract** Flaviviruses translate their genomes as multi-pass transmembrane proteins at the endoplasmic reticulum (ER) membrane. Here, we show that the ER membrane protein complex (EMC) is indispensable for the expression of viral polyproteins. We demonstrated that EMC was essential for accurate folding and post-translational stability rather than translation efficiency. Specifically, we revealed degradation of NS4A-NS4B, a region rich in transmembrane domains, in absence of EMC. Orthogonally, by serial passaging of virus on EMC-deficient cells, we identified two non-synonymous point mutations in NS4A and NS4B, which rescued viral replication. Finally, we showed a physical interaction between EMC and viral NS4B and that the NS4A-4B region adopts an aberrant topology in the absence of the EMC leading to degradation. Together, our data highlight how flaviviruses hijack the EMC for transmembrane protein biogenesis to achieve optimal expression of their polyproteins, which reinforces a role for the EMC in stabilizing challenging transmembrane proteins during synthesis.
DOI: https://doi.org/10.7554/eLife.48469.001

## Introduction

Flaviviruses such as dengue (DENV), Zika (ZIKV) or West Nile virus (WNV) express their genomes as a single multi-pass transmembrane proteins at the endoplasmic reticulum (ER) membrane. Subsequently, the polyprotein is cleaved by the viral NS2B-NS3 protease and the cellular signal peptidase into the individual structural (C, prM, E) and non-structural (NS1-5) proteins required for replication and assembly into new virions (*Neufeldt et al., 2018*). Similar to cellular transmembrane proteins, the viral polyprotein relies on the host cell machinery for targeting to the ER, translocation across the membrane and insertion of transmembrane domains (TMDs) into the lipid bilayer during its translation. These processes are facilitated by the signal recognition particle (SRP), the translocon-associated protein (TRAP) complex (also known as signal-sequence receptor complex), the Sec61 translocon and the signal peptidase complex. These cell components were all identified as essential host factors for flavivirus infection in genetic screens underscoring their importance for virus replication (*Krishnan et al., 2008*; *Marceau et al., 2016*; *Sessions et al., 2009*; *Zhang et al., 2016*). For most transmembrane proteins, recognition of a hydrophobic TMD or signal sequence by the SRP followed by transfer to the Sec61 translocation channel are sufficient for membrane targeting and accurate topogenesis (*Shao and Hegde, 2011*). Recently, it was shown that the ER membrane protein complex (EMC) plays a role in the insertion and/or stabilization of certain transmembrane proteins

(*Chitwood et al., 2018*; *Guna et al., 2018*; *Shurtleff et al., 2018*). Intriguingly, several genome-wide CRISPR knockout (KO) screens for flavivirus dependency factors showed strong enrichment of the EMC (*Ma et al., 2015*; *Marceau et al., 2016*; *Savidis et al., 2016*; *Zhang et al., 2016*). However, its role in the virus life cycle is not yet understood.

The EMC was originally discovered in a genetic screen for yeast mutants modulating the unfolded protein response (UPR) (*Jonikas et al., 2009*). Genetic interaction mapping and biochemical purification revealed a six-subunit core complex (EMC1-6) conserved in yeast and mammals (*Christianson et al., 2011*; *Jonikas et al., 2009*). Other than its more general association with UPR and ER-associated degradation (ERAD), the EMC was shown to interact co-translationally and enable biogenesis of a subset of multi-pass transmembrane proteins (*Shurtleff et al., 2018*). Biochemical characterization of select EMC client proteins, such as tail-anchored proteins and G protein-coupled receptors (GPCRs), revealed that the EMC is capable of inserting TMDs into the ER membrane (*Chitwood et al., 2018*; *Guna et al., 2018*). Here, we investigated how the mosquito-borne flaviviruses, such as DENV, ZIKV and WNV, depend on the EMC for their polyprotein biogenesis during infection.

## Results

To validate the enrichment of EMC components in the flavivirus CRISPR screens, we generated iso-genic EMC subunit KO cell lines in Huh7.5.1 and HEK293FT, two cell types that were used for the genetic screens (*Figure 1—figure supplement 1A*). Upon infection with DENV expressing Renilla luciferase (DENV-Luc), we measured a ~ 10,000 fold reduction in viral replication in Huh7.5.1 EMC1-

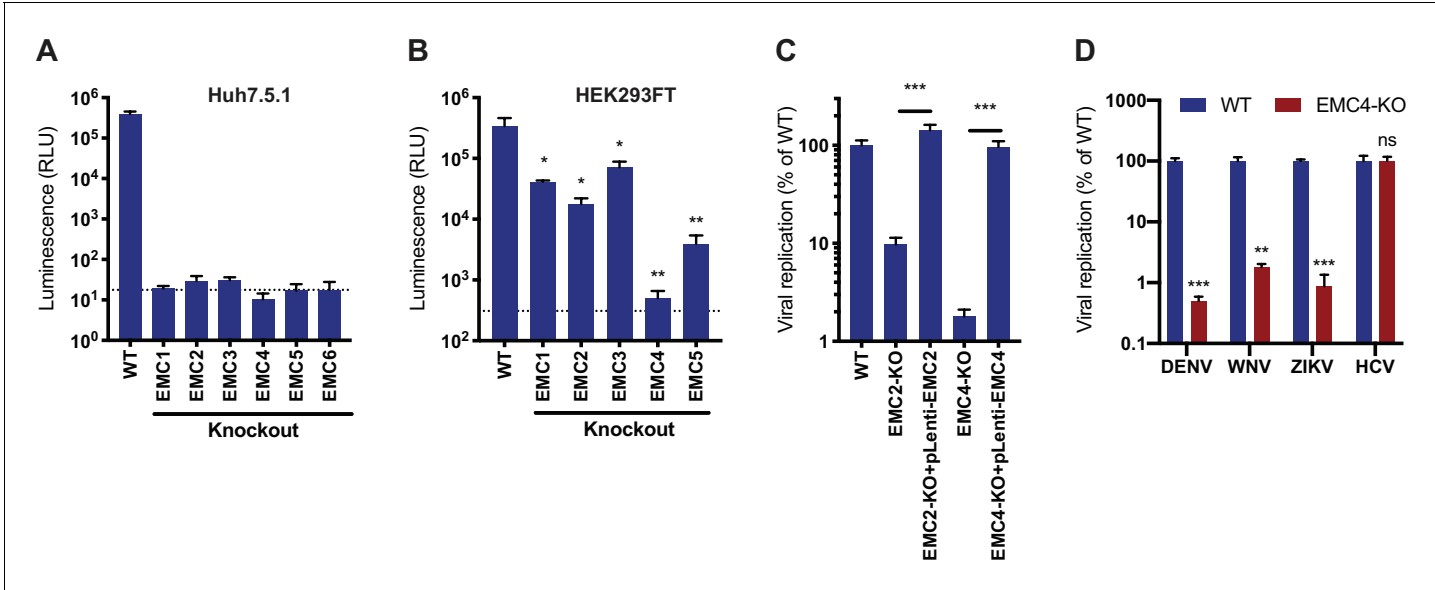

**Figure 1.** EMC is required for flavivirus infection. (**A**) DENV infection of WT and EMC subunit KO Huh7.5.1 cells. Cells were infected with DENV-Luc, harvested at 30hpi and luminescence was measured. Dotted line indicates background from uninfected control cells. (**B**) DENV infection of WT and EMC subunit KO HEK293FT cells. Cells were infected with DENV-Luc, harvested at 48hpi and luminescence was measured. Dotted line indicates background from uninfected control cells. (**C**) Replication of DENV in WT, EMC2- and EMC4-KO, and cDNA complemented KO HEK293FT cells. Cells were infected with DENV-Luc, harvested at 48hpi and luminescence was measured. (**D**) Quantitative RT-PCR of DENV, WNV, ZIKV and HCV RNA in WT or EMC4-KO Huh7.5.1 cells. Cells were infected with an moi of 0.5 for all viruses and harvested at 30hpi for ZIKV, 48hpi for DENV and WNV, and 72hpi for HCV. Viral RNA levels were normalized to 18S levels and data is displayed relative to the respective WT condition. In all figures, values are shown as mean of three biological replicates with standard deviation in (A)-(C), and as mean with standard error of the mean in (D). *t*-tests were performed to determine statistical significance and *p*-values are defined as ns = non significant, *<0.05, **<0.01, ***<0.001.

DOI: https://doi.org/10.7554/eLife.48469.002

The following figure supplement is available for figure 1:

**Figure supplement 1.** Genotyping and flavivirus infection of EMC KO cell lines.

DOI: https://doi.org/10.7554/eLife.48469.003

6 KO cells (*Figure 1A*) and a 10–1,000-fold reduction in HEK293FT EMC1-5 KO cells (*Figure 1B*). The differences in the extent of the phenotypes may be attributed to the presence of compensatory pathways or the degree of disruption of the entire protein complex between the two cell types upon knockout of a single subunit. The strong deficit in DENV replication in the absence of the EMC persisted over a 30–72 hr period (*Figure 1—figure supplement 1B*). Complementation of EMC2 and EMC4 KO cells with the respective cDNA completely restored levels of DENV infection (*Figure 1C* and *Figure 1—figure supplement 1C*). Lastly, we tested the effect of EMC4 KO on several flaviviruses and saw a significant decrease of viral RNA for DENV, WNV and ZIKV but not hepatitis C virus (HCV), which is a more distantly related member of the *Flaviviridae* family (*Figure 1D* and *Figure 1—figure supplement 1D*). This is congruent with results of an HCV CRISPR screen that did not identify the EMC as a critical host factor (*Marceau et al., 2016*). This finding highlights that there is a specific, evolutionarily conserved dependency on the EMC among mosquito-borne flaviviruses and not a universal disruption of trafficking to and/or translation or replication at the ER membrane, which would affect a wider range of viruses.

Next, we sought to determine at which step of the viral life cycle the EMC is required. As the EMC is important for the expression of a large set of transmembrane proteins, its disruption leads to an altered cell surface proteome, which may affect viral attachment to cells and/or entry. We measured effects on viral binding and uptake by quantifying the amount of viral RNA after virus incubation with WT and EMC4 KO cells at 2 and 6 hr post-infection (hpi), where we did not observe a difference (*Figure 2A*). Only at 24hpi, a timepoint when viral replication had occurred, there was a difference in RNA load evident between WT and EMC4 KO cells. Furthermore, we used a DENV replicon expressing luciferase, which bypasses entry and does not produce viral particles, thus allowing the analysis of the effect of EMC KO specifically on translation and replication. In this assay, we saw decreased luminescence in EMC4 KO cells starting at 12 hr post-electroporation suggesting that the EMC is important during or prior to viral replication (*Figure 2B*). We additionally performed the replicon assay in the presence of MK-0608, a nucleoside analogue that inhibits the viral polymerase (*Chen et al., 2015*), and observed that the luminescence signal in EMC4 KO cells was lower than in replication-inhibited WT cells suggesting that deletion of EMC has an effect on viral translation, which happens prior to replication (*Figure 2—figure supplement 1A*). To further dissect effects on translation or genome replication of DENV, we conducted an immunoblot timecourse series, where WT and EMC4 KO cells were infected at a high multiplicity of infection (moi = 32) and lysates were harvested at early timepoints post-infection (6-12hpi), when no or only little DENV replication had occurred. Viral protein expression was lower in EMC4 KO compared to WT cells as early as 6hpi further supporting a translation defect (*Figure 2C*). To rule out any contribution of low levels of replication to the observed phenotype, we also performed the immunoblot experiment in the presence of MK-0608 and still saw different amounts of DENV protein accumulation between WT and EMC4 KO cells indicating that the EMC is indeed important for viral protein expression (*Figure 2—figure supplement 1B*). Interestingly, while the EMC has been implicated in the stable biogenesis of transmembrane proteins, we found reduced expression of both TMD and non-TMD containing viral proteins. This suggests that the defect likely occurs at the polyprotein stage thus affecting all individual, cleaved proteins.

The expression of flaviviral proteins is a complex process that requires several steps. DENV translation is initiated in a cap-dependent manner, whereupon the single-stranded positive-sense RNA genome is translated into a single ~ 3400 amino acid long multi-pass transmembrane protein. This polyprotein is co- and post-translationally processed by host and viral proteases producing the individual structural and non-structural proteins. We first tested whether the defect occurs at the stage of polyprotein translation. We used ribosome profiling (Ribo-seq), which measures the density of ribosomes on a given mRNA species with high positional resolution, to compare translation efficiencies (TE) and successful full-length synthesis of both cellular and viral proteins in WT and EMC4 KO cells (*Ingolia et al., 2009*). Any changes in TE or premature ribosome stalling in EMC KO vs WT cells would be reflected by changes in ribosomal footprints (as measured by ribosome profiling) normalized to changes in mRNA abundance (as measured by RNA-seq). In the uninfected condition, we did not observe dramatic changes in TE for cellular mRNAs in EMC4 KO vs WT cells (*Figure 3A* and *Figure 3—source data 1*). This is in line with previously reported results using CRISPRi knockdown against EMC subunits suggesting that deletion of EMC minimally impacts the synthesis rates of its client proteins (*Shurtleff et al., 2018*). Similarly, we did not find any substantial differences in viral

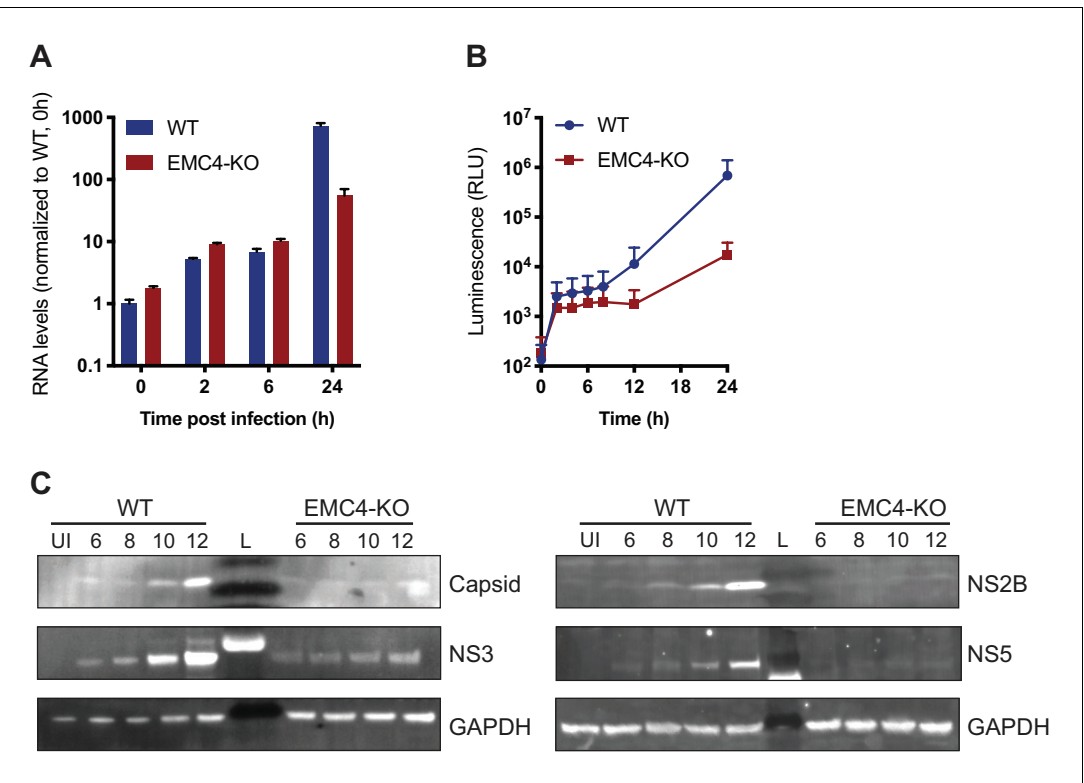

**Figure 2.** EMC is important for optimal viral protein expression. (A) Quantitative RT-PCR of DENV RNA from cell-bound and internalized virions in WT and EMC4-KO Huh7.5.1 cells. Cells were infected on ice with an moi of 30, incubated for 1 hr, then moved to a 37C incubator, and finally lysed at indicated timepoints. The data are from three biological replicates and shown as mean with standard error of the mean. (B) Luminescence of WT and EMC4-KO HEK293FT cells, which were electroporated with DENV replicon expressing Renilla luciferase and collected at different times post-electroporation. The data are from three biological replicates for each timepoint and shown as mean with standard deviation. (C) Immunoblot analysis of DENV proteins from infected WT and EMC4-KO HEK293FT cells at different timepoints post-infection (6/8/10/12 hr). Lysates were blotted for different viral proteins (Capsid, NS2B, NS3 and NS5) and GAPDH was used as loading control. UI = uninfected control; L = ladder.

DOI: https://doi.org/10.7554/eLife.48469.004

The following figure supplement is available for figure 2:

**Figure supplement 1.** EMC directly affects viral protein expression and not genome replication.

DOI: https://doi.org/10.7554/eLife.48469.005

TE in EMC4 KO cells relative to WT cells at 18 (0.43x) and 44hpi (2.17x) (*Figure 3B and C* and *Figure 3—source data 1*). Instead, we observed that DENV RNA was among the highest translated (top 0.1%) mRNAs in both WT and EMC4 KO cells (*Figure 3—figure supplement 1A*). The lower number of ribosomal footprints aligning to the viral genome in EMC4 KO cells was largely due to the reduced number of viral RNA copies present in the cells. Moreover, footprints were found to map throughout the viral genome in EMC4 KO cells, indicating that synthesis of full-length polyprotein still occurred (*Figure 3—figure supplement 1B*). Therefore, we concluded that translation of DENV polyproteins is not strongly impacted by loss of EMC, which led us to explore potential post-translational defects.

Proper insertion of TMDs and correct folding are required for stable expression of transmembrane proteins, while misfolded proteins are generally targeted to the proteasome for degradation via the ERAD pathway (*Olzmann et al., 2013*). To probe whether an increased fraction of viral proteins is degraded by the proteasome in EMC deficient cells, we infected WT and EMC4 KO cells with DENV in the presence of Bortezomib (BZ), a proteasome inhibitor, thus preventing removal of misfolded proteins from the cells. Indeed, BZ treatment led to an increase of detectable viral

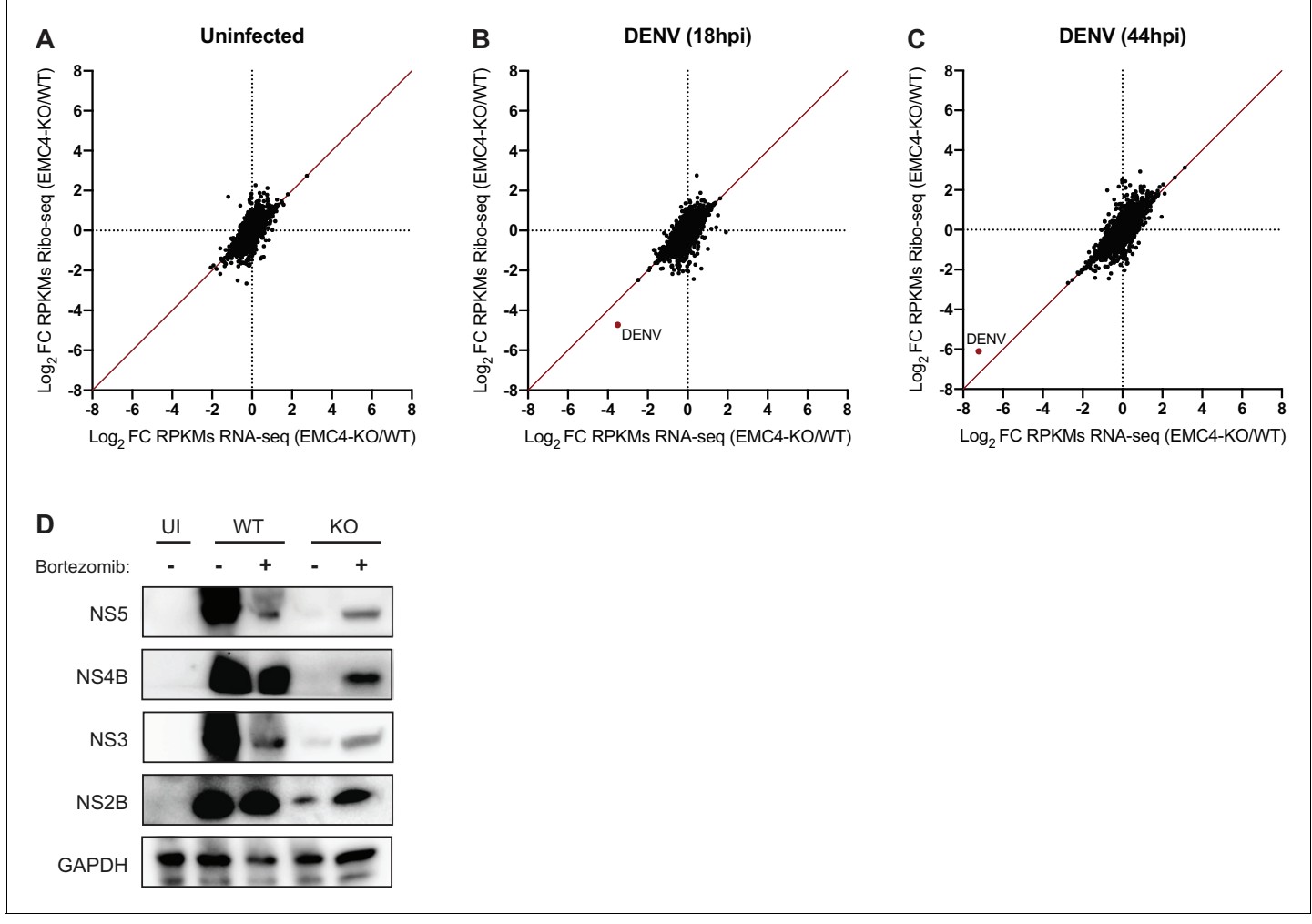

**Figure 3.** EMC knockout does not largely affect viral translation efficiency but leads to post-translational protein degradation. (A) Ribosome Profiling (Ribo-seq) and RNA-seq were performed in uninfected WT and EMC4-KO HEK293FT cells to measure changes in translation efficiency (TE). Cycloheximide was added to stop translation and total RNA was isolated. RNA was treated with RNaseI and ribosome-protected footprints were purified, reverse transcribed and amplified for next-generation sequencing. Log2 fold-change (FC) of reads per kb of transcript, per million mapped reads (RPKMs) between EMC4-KO and WT cells for Ribosome Profiling and RNA-seq are displayed on the y- and x-axis, respectively. Each dot represents one RNA transcript. Fold-change in TE for a given transcript is defined as the change in Ribosome Profiling RPKMs normalized to the change in RNA-seq RPKMs. The red line represents fold-change of TE = 1. The Ribosome profiling experiments were performed once for each condition. (B) Ribosome Profiling (Ribo-seq) and RNA-seq were performed in DENV-infected WT and EMC4-KO HEK293FT cells 18 hr post-infection to measure changes in translation efficiency (TE). The DENV transcript is highlighted in red. The red line represents fold-change of TE = 1. (C) Ribosome Profiling (Ribo-seq) and RNA-seq were performed in DENV-infected WT and EMC4-KO HEK293FT cells 44 hr post-infection to measure changes in translation efficiency (TE). The DENV transcript is highlighted in red. The red line represents fold-change of TE = 1. (D) Viral protein expression from DENV infected WT and EMC4-KO HEK293FT cells with or without addition of bortezomib (50 nM), a proteasome inhibitor. Cells were harvested at 18hpi and lysates were analyzed by immunoblot using antibodies against dengue proteins. GAPDH was used as loading control. UI = uninfected. .
DOI: https://doi.org/10.7554/eLife.48469.006

The following source data and figure supplements are available for figure 3:

**Source data 1.** Ribosome Profiling data including read counts and fold changes for ribosome footprints and RNA-seq used to generate *Figure 3A–C*.
DOI: https://doi.org/10.7554/eLife.48469.009
**Figure supplement 1.** DENV efficiently translates and produces full-length polyproteins in EMC KO cells.
DOI: https://doi.org/10.7554/eLife.48469.007
**Figure supplement 2.** Bortezomib-recovered viral proteins do not restore viral replication.
DOI: https://doi.org/10.7554/eLife.48469.008

proteins in EMC4 KO cells but not WT cells (*Figure 3D*). Addition of BZ, however, did not restore DENV replication arguing that the proteasome-targeted proteins are non-functional (*Figure 3—figure supplement 2*). Together, this suggests that in the absence of EMC viral proteins are still translated but are unable to achieve a stable conformation resulting in their degradation and inability to establish efficient replication.

For cellular multi-pass client proteins, such as transporters and GPCRs, the EMC co-translationally engages with specific TMDs in order to support insertion into the membrane and allow stable expression of the entire transmembrane protein (*Chitwood et al., 2018*; *Shurtleff et al., 2018*). We used two orthogonal methods, a fluorescence-based protein stability assay and viral evolution, to examine which region of the viral polyprotein requires the EMC for successful expression. First, we utilized a dual-color fluorescent reporter as previously described (*Chitwood et al., 2018*). The C terminus of different segments of the DENV polyprotein was fused to eGFP followed by a viral T2A sequence and mCherry (G2C) (*Figure 4A*). Translation of the fusion constructs generates two protein products at a 1:1 ratio due to peptide bond skipping by the ribosome at the T2A site: 1) a polyprotein segment fused to eGFP and 2) mCherry. Any post-translational instability of the eGFP fusion protein due to misfolding results in an eGFP:mCherry ratio of less than 1. As expected, the cytosolic G2C-only construct displayed a 1:1 ratio in both WT and EMC4 KO cells, while the $\beta_1$-adrenergic receptor (ADRB1) fused to G2C exhibited drastic destabilization in EMC4 KO compared to WT cells as was previously reported (*Figure 4B*) (*Chitwood et al., 2018*). Applying this approach to different DENV polyprotein G2C fusion constructs, we observed that the segments containing only the structural proteins (capsid, prM and E) or NS1-3 were not destabilized in EMC4 KO Huh7.5.1 cells (*Figure 4B*). By contrast, constructs encoding both NS4A and NS4B (but not either of them alone) showed reduced stability in EMC4 KO cells (*Figure 4B*). Similar results were obtained in HEK293FT (*Figure 4—figure supplement 1*). NS4A has 3 TMDs with the third (termed 2K peptide) being highly hydrophobic and serving as a signal sequence for the translocation of NS4B into the ER membrane (*Lin et al., 1993*). Subsequently, the NS4A-2K junction is cleaved by the viral protease and the 2K-NS4B junction by the signal peptidase. As the 2K-targeted NS4B-5-G2C construct did not display a stability defect in EMC deficient cells, the results suggest that EMC mediates stable expression by first engaging within NS4A prior to the 2K peptide and that a defect in NS4A is extended to the NS4B region. Stability of NS4A-4B-G2C was restored to near WT levels by complementation of EMC4 KO cells with EMC4 cDNA (*Figure 4C*). Moreover, treatment of EMC4 KO cells with BZ increased the eGFP:mCherry ratio of NS4A-4B-G2C in EMC4 KO cells indicating that this region of the polyprotein is misfolded and targeted for degradation in the absence of EMC (*Figure 4D*). Therefore, we linked the DENV replication phenotype in EMC deficient cells to a specific molecular defect in the NS4A-4B region of the polyprotein.

In a second, orthogonal approach, we performed viral adaptation to evolve EMC-independent DENV mutants by passaging the virus consecutively on EMC4 KO Huh7.5.1 cells every 3–4 days. After 20–22 passages we observed cell death in 3 out of 6 independent adaptation experiments. Sequencing of RNA extracted from supernatant and alignment to the WT16681 DENV genome revealed two non-synonymous point mutations in NS4A and NS4B, which were conserved across all three isolates (*Figure 5A*). The A6665G mutation led to a tyrosine to cysteine and the A7558T mutation to an asparagine to tyrosine substitution. The majority of other SNPs was silent (*Figure 5—source data 1*). A6665G is predicted to be inside a TMD helix encoding region of NS4A suggesting that this TMD relies on the EMC for proper ER membrane insertion (*Miller et al., 2007*). Only one of 1,359 DENV serotype two genomes from the NCBI database has a G at this position encoding cysteine, while other sequences all encode tyrosine. This evolutionary conservation suggests that this residue indeed has an important function. Similarly, the frequency of A at position 7558 is very conserved with 99.6%, and there is no T encoding tyrosine in the 1359 sequenced strains. Interestingly, A7558T resides in a non-TMD region at the end of NS4B, thus likely not supporting membrane insertion but folding in a different manner (*Miller et al., 2006*). For all three EMC adapted isolates, replication levels in EMC4 KO cells were similar to levels in WT cells, while WT16681 DENV displayed a ~ 100 fold reduction in viral RNA in EMC4 KO relative to WT cells (*Figure 5—figure supplement 1A*). The same effect was observed in a 72 hr viral growth curve (*Figure 5B*). The adapted virus appeared less fit than the WT DENV at earlier timepoints but reached similar levels of replication at 72 hpi suggesting that the adaptive mutations did not cause dramatic growth defects.

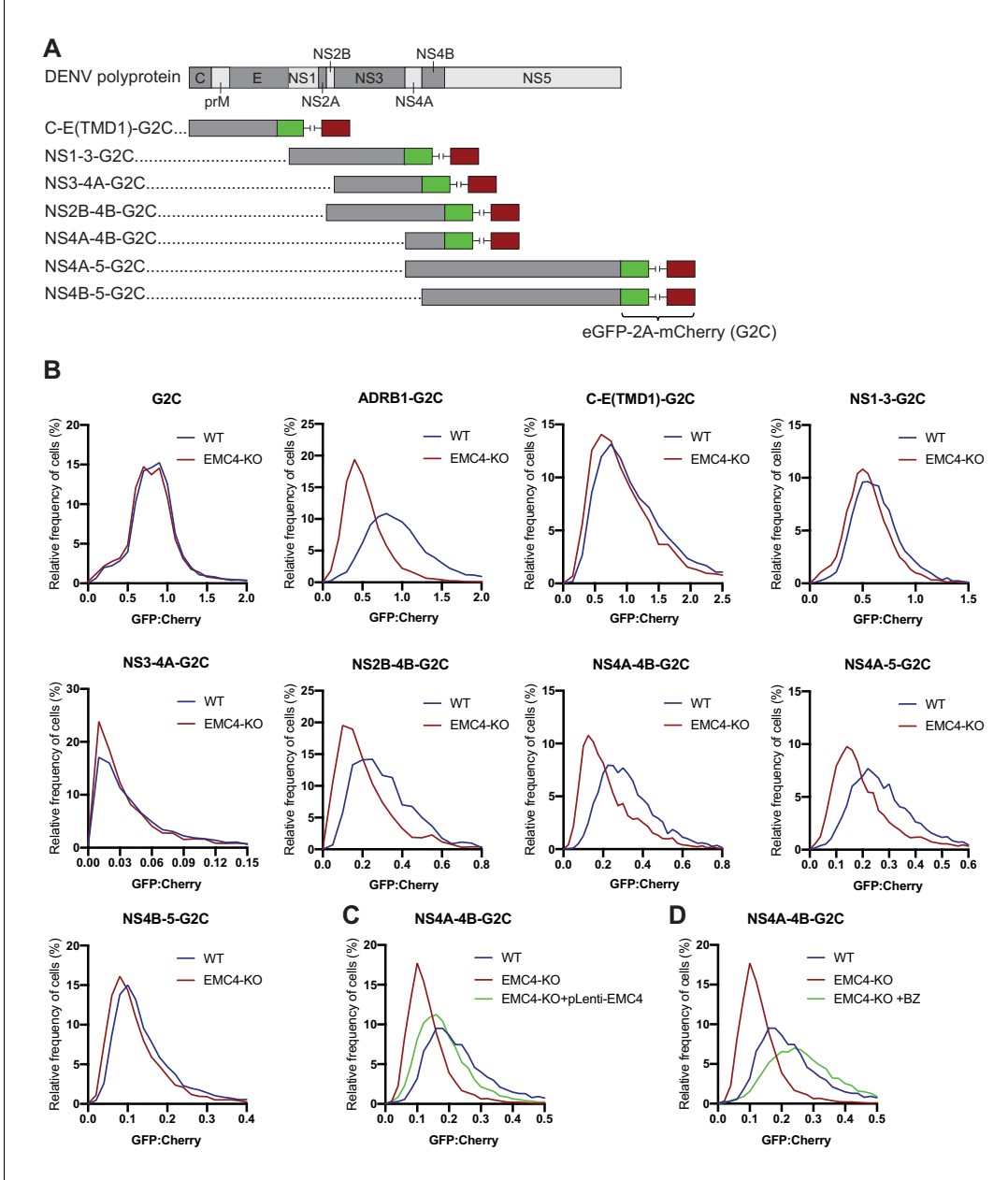

**Figure 4.** EMC is required for proper biogenesis of the NS4A-4B region of the DENV polyprotein. (**A**) Overview of dual-fluorescence reporter constructs of DENV polyprotein for analysis of protein stability. Segments of the DENV genome encoding parts of the polyprotein were cloned with a C-terminal fusion to eGFP-T2A-mCherry (G2C). The T2A peptide leads to peptide bond skipping during translation resulting in two protein products, DENV protein fused to eGFP and mCherry. The GFP:mCherry fluorescence ratio reflects changes in DENV protein stability. (**B**) GFP:mCherry fluorescence ratio for G2C constructs in WT and EMC4-KO Huh7.5.1 cells. Cells were transfected with G2C construct and analyzed by flow cytometry after 24–48 hr. For each individual cell the GFP:mCherry ratio was calculated and the fluorescence ratios are depicted as histograms. ADRB1=β1-adrenergic receptor. (**C**) Measurement of GFP:mCherry fluorescence ratios of NS4A-4B-G2C construct in WT, EMC4-KO and EMC4 cDNA complemented EMC4-KO HEK293FT cells. (**D**) Measurement of GFP:mCherry fluorescence ratios of NS4A-4B-G2C construct in WT, EMC4-KO and bortezomib-treated EMC4-KO HEK293FT cells. Note that (**C**) and (**D**) use same WT and EMC4-KO data as experiments were performed in parallel.

DOI: https://doi.org/10.7554/eLife.48469.010

The following figure supplement is available for figure 4:

**Figure supplement 1.** EMC is required for proper biogenesis of the NS4A-4B region of the DENV polyprotein in HEK293 cells.

DOI: https://doi.org/10.7554/eLife.48469.011

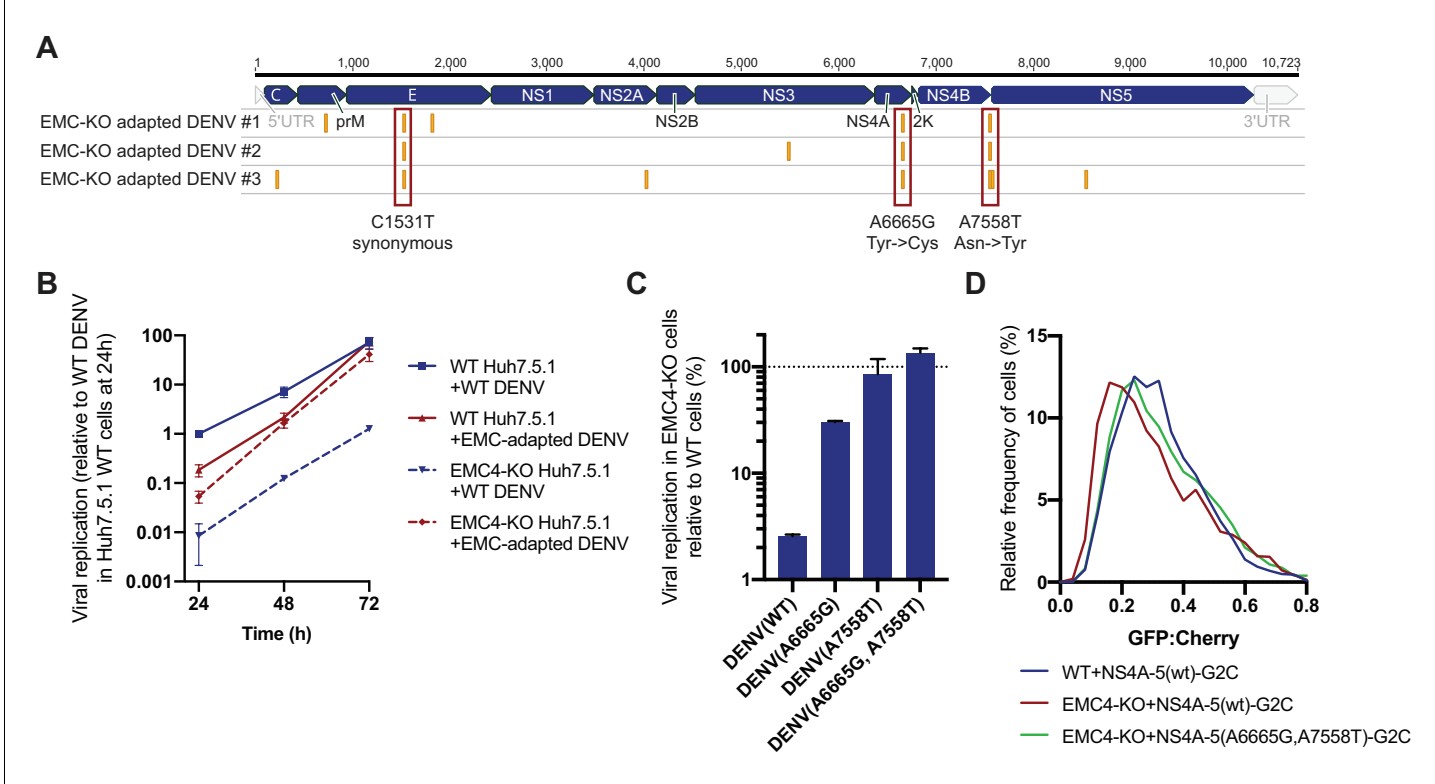

**Figure 5.** Adaptation of DENV to EMC KO cells reveals non-synonymous point mutations in NS4A and NS4B. (**A**) Identification of SNPs in EMC4-KO adapted DENV relative to DENV 16681 reference genome by next-generation sequencing of supernatant-extracted viral RNA. DENV was serially passaged on EMC4 KO Huh7.5.1 cells every 3–4 days. After 20–22 passages supernatant was collected and RNA extracted from three independent passaging experiment. Truseq total RNA sequencing libraries were generated and reads were mapped to the reference genome. Mutations common across all three replicates are highlighted by red boxes and their nucleotide and amino acid changes are shown below. A table of all identified SNPs can be found in *Figure 5—source data 1*). (**B**) Viral growth curves of WT and EMC-adapted DENV in WT and EMC4 KO cells measured by quantitative RT-PCR. Cells were infected with WT or EMC-adapted virus at an moi of 0.5 and harvested at indicated timepoints. The data are from three biological replicates and shown as mean with standard error of the mean. (**C**) Viral replication of WT DENV-Luc or mutant DENV-Luc (containing A6665G and A7558T either individually or in tandem) in EMC4 KO cells relative to WT cells. Lysates were collected 72hpi and luminescence was measured. The data are from three biological replicates and shown as mean with standard deviation. (**D**) Measurement of GFP:mCherry fluorescence ratios of WT NS4A-5-G2C and NS4A-5(A6665G,A7558T)-G2C constructs in WT and EMC4-KO Huh7.5.1.

DOI: https://doi.org/10.7554/eLife.48469.012

The following source data and figure supplement are available for figure 5:

**Source data 1.** Identified SNPs in EMC KO adapted DENV isolates relative to the DENV 16681 reference genome.
DOI: https://doi.org/10.7554/eLife.48469.013

**Figure supplement 1.** DENV containing EMC KO adapted mutations efficiently replicates in EMC deficient cells.
DOI: https://doi.org/10.7554/eLife.48469.014

To dissect whether both mutations are required for optimal replication in EMC KO background, we introduced them individually or in tandem into the DENV-Luc infectious clone and generated virus. While A6665G and A7558T individually restored DENV replication in EMC4 KO cells to 30% and 86% of WT levels, respectively, the double mutant displayed complete rescue of viral growth in EMC KO cells (*Figure 5C*). Similarly, we measured ~ 10 and 100-fold higher replication levels for the single and double mutant DENV-Luc, respectively, compared to WT DENV-Luc in EMC KO cells over a 72 hr period, while replication in WT cells was comparable (*Figure 5—figure supplement 1B*). We also introduced the two adaptive mutations into NS4A-5-G2C, which showed increased stability in EMC4 KO cells compared to the WT construct thus reverting the molecular destabilization pheno-type (*Figure 5D*). Remarkably, the G2C fluorescence assay and viral adaptation independently pin-pointed NS4A-4B of the polyprotein as the region, which requires the EMC. The results of the two experiments suggest that a defect in the stable insertion of the NS4A TMD2 could lead to an

aberrant topology/instability of NS4B. By contrast, polyprotein segments starting with 2K-NS4B were stably expressed in EMC KO cells as was shown using NS4B-5-G2C construct (*Figure 4A*). Therefore, the EMC is required for correct insertion, processing and/or folding of the NS4A-4B region.

To probe further how absence of EMC causes failure of DENV replication, we transfected a sub-genomic NS2B-4B construct (containing functional NS2B-NS3 for polyprotein cleavage) with a C-terminal fusion to eGFP into WT and EMC4 KO cells and immunoblotted against the different processed products. While there was no notable difference in expression levels between the higher molecular weight intermediates (NS2B-NS3-NS4A-NS4B-eGFP or NS4A-NS4B-eGFP), we observed reduced accumulation of processed NS4B-eGFP in EMC4 KO cells (*Figure 6A*). BZ treatment indicated rapid degradation of cleaved NS4B-eGFP in EMC deficient cells. Furthermore, if the EMC acts to insert or stabilize NS4A-4B, we hypothesized that direct binding of NS4B to EMC post polyprotein-processing should be detectable in WT cells. We performed a co-immunoprecipitation using FLAG-tagged EMC4 and detected NS4B but not prM or NS5 in the pulldown (*Figure 6B*). This supports the finding that EMC physically engages with NS4A-4B to facilitate its biogenesis and stable expression. Interestingly, we also detected physical interaction between the EMC and NS4B of the adapted DENV (*Figure 6C*). This suggests that the two amino acid substitutions in the NS4A and NS4B viral proteins do not ablate the signal triggering EMC engagement. The mutations may rather allow the virus to maintain proper folding and stability in the absence of the EMC.

Finally, we examined the topology of the NS4A-4B transmembrane protein region in WT and EMC deficient cells using a protease protection assay. We generated different constructs, in which eGFP was C-terminally fused after different TMDs that had been determined experimentally (*Figure 6D*) (*Miller et al., 2006*). Transfected cells were left untreated or treated with digitonin and proteinase K followed by immunoblot. In the case of luminal localization, eGFP is protected against protease digestion, while cytoplasmic eGFP is degraded. Our results showed that in WT cells eGFP fused after NS4B TMD1 and TMD4 is luminal, which is consistent with the previously determined topology of NS4B (*Figure 6D*) (*Miller et al., 2006*). By contrast, we did not observe luminal localization for NS4B TMD1 and TMD4 in EMC4 KO cells, which indicates that EMC is required for the achieving the correct topology and thus stable expression of NS4B. We additionally probed the topology of NS4A using a deglycosylation assay similar to a previous study (*Miller et al., 2007*). We introduced an opsin tag containing a glycan acceptor site downstream of NS4A TMD2, which had been determined to be in the ER lumen and should therefore be glycosylated. We observed efficient glycosylation in WT cells (as expected) but very poor glycosylation in EMC4 KO cells, indicating an aberrant topology (*Figure 6E*). Moreover, we sought to test whether the adaptive mutations restore the topology of NS4A-4B or allow the virus to replicate with a different topology in the absence of the EMC. Since a topological defect was observed early in NS4A-4B and membrane translocation occurs co-translationally, we focused on the A6665G mutation, which is located in TMD2 of NS4A, rather than the A7558T mutation near the C-terminus of NS4B. As the A6665G mutation led to a significant increase in replication in EMC deficient cells, we would expect any rescue of correct topology to be visible. However, no increase of glycosylation was observed in EMC4 KO cells using the NS4A-TMD2-opsin construct containing the adaptive mutation (*Figure 6E*). Therefore, we conclude that the adaptive mutation in NS4A does not immediately restore the correct topology but instead allows the virus to replicate with NS4A-4B assuming an aberrant topology. However, we cannot exclude that the adaptive mutations, which were selected by passaging full virus, only restore the wild-type topology in the context of full-length viral polyprotein, for example through interactions with other viral proteins.

## Discussion

Based on our results and previous work (*Chitwood and Hegde, 2019*; *Shurtleff et al., 2018*) we propose the following model for the role of the EMC in flavivirus infection. During translation of the polyprotein the EMC engages with TMDs in NS4A and NS4B to ensure their accurate topology, correct folding and stable expression (*Figure 6F*). In the absence of the EMC, TMDs in the NS4A-4B region are not accurately or stably integrated in the ER membrane leading to misfolding and degradation of processed NS4B (and potentially also NS4A). Based on the molecular weight of the detected bands in the protease protection assay (*Figure 6D*), we expect that cleavage between 2K

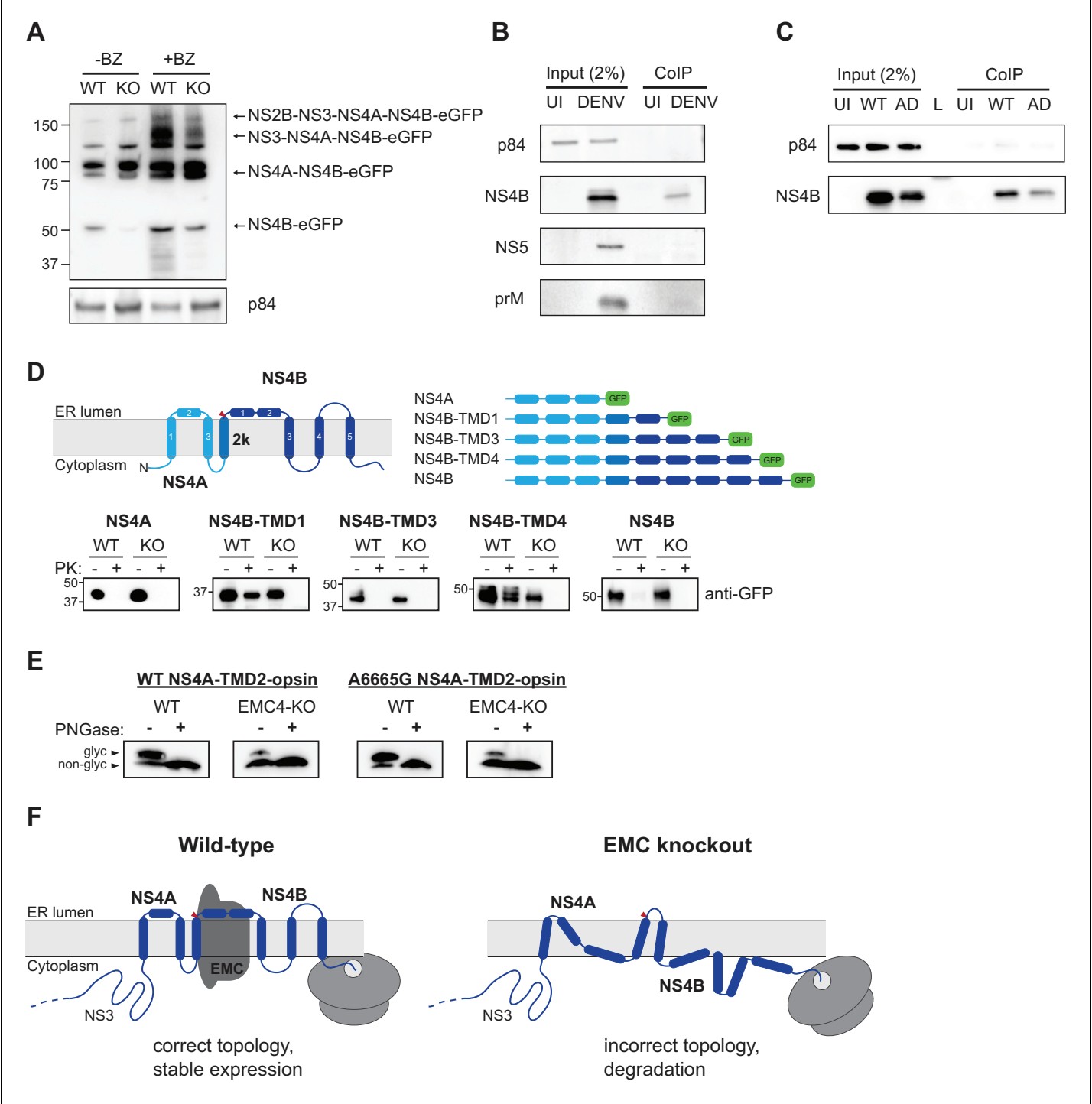

**Figure 6.** EMC physically interacts with DENV NS4B post-cleavage and knockout of EMC leads to an aberrant topology of NS4A and NS4B. (**A**) Immunoblot for processed products of transfected NS2B-4B-GFP construct in WT and EMC4 KO cells in presence or absence of BZ. Anti-GFP antibody was used to detect processed viral proteins and p84 was used as loading control. (**B**) Co-immunoprecipitation of EMC4-FLAG from lysates of uninfected (UI) or DENV infected HEK293FT EMC4-FLAG cells. Cells were harvested after 96 hr and lysates were incubated with anti-FLAG magnetic beads. Eluates were immunoblotted for different viral proteins (prM, NS4B, NS5). p84 was used as loading and non-specific binding control. (**C**) Co-immunoprecipitation of EMC4-FLAG from lysates of uninfected cells (UI), cells infected with WT DENV (WT) or infected with EMC adapted DENV (AD). (**D**) Protease protection assay for different NS4A-4B-eGFP fusion constructs. Experimentally determined topology according to *Miller et al. (2006)*; *Miller et al. (2007)* and generated constructs are displayed. Different shades of blue indicate NS4A, 2K and NS4B in the model and red triangle points to signal peptidase cleavage site. Constructs were transfected into WT or EMC4-KO cells. After 24–48 h cell pellets were resuspended in buffer with

*Figure 6 continued on next page*

*Figure 6 continued*

digitonin, and subsequently treated with proteinase K (PK) or left untreated. Finally, samples were lysed and immunoblotted using anti-GFP antibody. (E) Deglycosylation assay using PNGase F to probe luminal localization of NS4A TMD2 using constructs containing the NS4A WT sequence or A6665G mutation. Opsin tags containing a N-K-T sequon were inserted downstream of NS4A TMD2. WT or EMC4-KO cells were transfected with WT or mutant construct and harvested after 24 hr. Lysates were treated with PNGase F to remove N-linked oligosaccharides or left untreated following analysis by immunoblot using anti-FLAG antibody. Upper bands represent NS4A with glycosylated and thus luminally localized opsin tag while lower bands are non-glycosylated forms. (F) Possible model for the role of EMC in flavivirus infection. In infected WT cells, the EMC co-translationally engages with NS4A and NS4B TMDs to facilitate or maintain correct insertion and topology of TMDs , thus ensuring stable expression. In the absence of the EMC, an aberrant topology is obtained during biogenesis, leading to misfolding and degradation of the NS4A-4B viral proteins.

DOI: https://doi.org/10.7554/eLife.48469.015

and NS4B by the signal peptidase still occurs in EMC deficient cells, which is consistent with previously observed processing of dengue NS4A-2K-NS4B (*Cahour et al., 1992*). This suggests that the N-terminus of NS4B is still localized to the ER lumen (at least while being translated) in EMC KO cells. However, unlike in WT cells, where the TMD1 and TMD4 of NS4B are luminal, their termini are localized to the cytoplasmic site of the ER membrane in EMC KO cells highlighting that EMC is required for achieving the correct topology of NS4B. Our data suggests that EMC does not affect ER targeting and translocation of 2K but it still remains an open question whether EMC facilitates proper insertion of TMDs into the membrane or if it is required for maintaining the correct topology in a chaperone-like function post membrane integration.

NS4A and NS4B fulfill crucial functions for viral replication. NS4A induces ER membrane rearrangements important for the formation of viral replication complexes (*Miller et al., 2007*; *Roosendaal et al., 2006*) while NS4B interacts with the NS3 helicase and also suppresses the cellular interferon response (*Muñoz-Jordán et al., 2005*; *Zou et al., 2015*). Additionally, it was demonstrated that the NS4A-2K-NS4B precursor interacts with NS1, important for viral replication (*Płaszczyca et al., 2019*).

Recently, another study found that biogenesis of DENV and ZIKV NS4A and NS4B is dependent on the EMC (*Lin et al., 2019*). While Lin et al. used ectopic expression of individual viral proteins to identify NS4A and NS4B as EMC client proteins, we performed a fluorescent based stability assay expressing different fragments of the viral polyprotein leading to similar conclusions. However, we only observed destabilization when NS4B was preceded by NS4A arguing that the NS4A-4B intermediate triggers an EMC-dependent process. Both studies showed that the EMC physically interacts with NS4B to facilitate stable expression and that, in absence of EMC, NS4B is misfolded and targeted for degradation as demonstrated by proteasomal inhibition. Lin et al. showed that NS4B is still membrane associated in EMC deficient cells, while we provided further molecular insights that NS4A-4B assumes an aberrant topology at the ER membrane. Furthermore, bioinformatic prediction of NS4B transmembrane regions revealed that the two N-terminal domains are only weakly hydrophobic, which could explain the dependence of NS4B on the EMC for stable expression (*Lin et al., 2019*). This is consistent with the systematic analysis of cellular EMC client proteins, which showed an underrepresentation of hydrophobic and an overrepresentation of charged and aromatic amino acids within their TMDs compared to all TMDs in the proteome (*Shurtleff et al., 2018*). Consequently, Lin et al. demonstrated that substitution of all polar and charged residues inside TMD1 and TMD2 of NS4B to hydrophobic leucines led to complete rescue of ectopic NS4B expression in EMC depleted cells. However, it remains unclear whether these substitutions would allow for viral replication in the full polyprotein context. By contrast, our EMC KO adapted mutations do not reside in TMD1 and TMD2 of NS4B but in NS4A TMD2 and near the C-terminus of NS4B, and were shown to increase NS4A-4B expression levels as well as restore viral replication. Overall, both studies are complementary and in large agreement, and together provide a detailed picture of how the EMC ensures stable expression of two viral multi-pass transmembrane proteins.

Interestingly, we found that the A6665G mutation inside NS4A TMD2 does not restore the wild-type topology. The A6665G mutation also arose during DENV adaptation to cell lines deficient in the oligosaccharyltransferase (OST) complex, another critical host factor inside the ER membrane (*Marceau et al., 2016*; *Puschnik et al., 2017*). The OST complex has been shown to be important for RNA genome replication in a catalytically independent function and was suggested to act as structural scaffold for viral replication complexes. It is thus possible that the tyrosine to cysteine

substitution generally modulates ER membrane integration of NS4A-4B or exposes a binding interface important for replication complex formation, thus likely rescuing replication without restoring the normal topology. Since membrane proteins are co-translationally translocated across and inserted into the membrane, it is unlikely that the A7558T mutation downstream of the apparent topology defect restores proper membrane integration. Possibly, the substituting tyrosine may stabilize the cytosolic orientation of the C-terminus through expansion of the 'aromatic belt' (*Granseth et al., 2005*) to allow for efficient viral protease cleavage or facilitate other critical protein-protein interactions.

The isolation of EMC-independent DENV mutants additionally enables drug screening for EMC modulators by testing compounds against WT DENV-Luc and then counter-screening hits with EMC-adapted virus. EMC inhibitors may be a useful chemical tool to study EMC function and also provide a potential antiviral therapeutic against flaviviruses. Host-directed antivirals may have the advantage of a broader activity against multiple viruses and a higher barrier of drug resistance. Only 50% of the adaptation experiments yielded escape mutants containing two mutations after > 20 passages. For comparison, for direct-acting antivirals (e.g. viral polymerase inhibitors), one point mutation is often sufficient to overcome the mechanism of action (*Beaucourt and Vignuzzi, 2014*).

In addition to characterizing the role of the EMC in the viral life cycle, our data also help to further elucidate its cellular function. The emerging role of the EMC to facilitate membrane protein insertion and biogenesis has been implicated for several cellular transmembrane proteins (*Chitwood et al., 2018*; *Coelho et al., 2019*; *Guna et al., 2018*; *Shurtleff et al., 2018*; *Volkmar et al., 2019*). For a subset of tail-anchored proteins, the EMC inserts the sole C-terminal TMD after translation is completed in the cytosol (*Guna et al., 2018*). Moreover, the EMC co-translationally inserts the first N-terminal TMD of many GPCRs with an $N_{exo}$ topology (N terminus in the ER lumen) (*Chitwood et al., 2018*). By contrast, we found EMC-dependency within DENV NS4A-4B, a region with several TMDs, which are not close to the termini of the polyprotein. Moreover, as NS4A is already targeted to and inserted into the ER membrane, it argues against an insertase mechanism (as shown for tail-anchored proteins and GPCRs) to ensure NS4B expression. This provides experimental evidence that the EMC is not only capable of supporting stable expression of TMDs near the N and C termini but also of internal TMDs, arguing that the EMC may directly facilitate the biogenesis of a wide variety of implicated transmembrane proteins beyond tail-anchored proteins and GPCRs as suggested previously (*Shurtleff et al., 2018*). It is noteworthy that the DENV polyprotein is distinct from human transmembrane proteins in that it contains multiple signal peptidase cleavage sites along the polyprotein, including at the 2K/NS4B site between NS4A and NS4B, which removes the 2K peptide acting as signal sequence for NS4B (*Lin et al., 1993*; *Miller et al., 2006*). Interestingly, it was shown for ADRB1 that addition of a signal sequence or an $N_{cyt}$ TMD upstream of the first $N_{exo}$ TMD makes the EMC dispensable for biogenesis (*Chitwood et al., 2018*), whereas NS4B is EMC-dependent despite the upstream 2K signal peptide within the polyprotein. This observation suggests that EMC-mediated integration/stabilization of TMDs may be context-dependent. Generally, for both cellular multi-pass transmembrane proteins and the DENV polyprotein, only one or a subset of TMDs seem to require the EMC for accurate topology, while the other TMDs rely on the Sec61 translocon. The exact interplay between Sec61 and the EMC during transmembrane protein biogenesis warrants further investigation. The simultaneous dependency of flaviviruses on both Sec61 and EMC as highlighted by the CRISPR KO screens could allow an in-depth analysis of the sequence and TMD features that may determine membrane insertion via Sec61 or EMC.

Lastly, the EMC was originally discovered to be associated with the UPR (*Jonikas et al., 2009*), and upregulation of UPR during flavivirus infection has been reported (*Peña and Harris, 2011*; *Reid et al., 2018*; *Su et al., 2002*). It is possible that competition between viral polyproteins and cellular proteins for the machinery important for proper transmembrane protein expression (such as EMC, OST complex and ERAD components) is linked to this phenomenon. The induction of ER stress can further trigger autophagy, which has also been shown to be beneficial for flavivirus infection (*Abernathy et al., 2019*; *Lee et al., 2018*; *Senft and Ronai, 2015*). Therefore, hijacking the EMC may have direct and indirect implications for flavivirus replication.

Overall, our study gives detailed insights into how flaviviruses exploit the EMC to support proper polyprotein expression, and we provide further evidence for the cellular function of the EMC in ensuring accurate transmembrane protein topology by utilizing DENV as a cell biological tool.

# Materials and methods

## Key resources table

| Reagent type (species) or resource | Designation | Source or reference | Identifiers | Additional information |
|---|---|---|---|---|
| Cell line (human) | HEK293FT | ThermoFisher | R70007 | |
| Cell line (human) | EMC1-KO HEK293FT | This study | | See Material and Methods, 'Generation of knockout cell lines' |
| Cell line (human) | EMC2-KO HEK293FT | This study | | See Material and Methods, 'Generation of knockout cell lines' |
| Cell line (human) | EMC3-KO HEK293FT | This study | | See Material and Methods, 'Generation of knockout cell lines' |
| Cell line (human) | EMC4-KO HEK293FT | This study | | See Material and Methods, 'Generation of knockout cell lines' |
| Cell line (human) | EMC5-KO HEK293FT | This study | | See Material and Methods, 'Generation of knockout cell lines' |
| Cell line (human) | Huh7.5.1 | PMID: 15939869 | RRID: CVCL_E049 | |
| Cell line (human) | EMC1-KO HUH7.5.1 | This study | | See Material and Methods, 'Generation of knockout cell lines' |
| Cell line (human) | EMC2-KO HUH7.5.1 | This study | | See Material and Methods, 'Generation of knockout cell lines' |
| Cell line (human) | EMC3-KO HUH7.5.1 | This study | | See Material and Methods, 'Generation of knockout cell lines' |

*Continued on next page*

*Continued*

| Reagent type (species) or resource | Designation | Source or reference | Identifiers | Additional information |
|---|---|---|---|---|
| Cell line (human) | EMC4-KO HUH7.5.1 | This study | | See Material and Methods, 'Generation of knockout cell lines' |
| Cell line (human) | EMC5-KO HUH7.5.1 | This study | | See Material and Methods, 'Generation of knockout cell lines' |
| Cell line (human) | EMC6-KO HUH7.5.1 | This study | | See Material and Methods, 'Generation of knockout cell lines' |
| Biological sample (virus) | DENV serotype 2 strain 16681 | PMID: 9143286 | NCBI Reference Sequence: NC_001474.2 | Generated from infectious clone obtained from Karla Kirkegaard |
| Biological sample (virus) | DENV-Luc | PMID: 27383987 | | |
| Biological sample (virus) | WNV Kunjin strain CH16532 | Other | GenBank: JX276662.1 | Source: John F. Anderson |
| Biological sample (virus) | ZIKV PRVABC59 | BEI Resources | NR-50240 | |
| Biological sample (virus) | HCV JFH1 | PMID: 11424123 | GenBank: AB047639.1 | |
| Antibody | P84 (mouse, monoclonal) | Genetex | GTX70220 | 1:2000 |
| Antibody | GAPDH (mouse, monoclonal) | SCBT | sc-32233 | 1:2000 |
| Antibody | EMC2/TTC35 (rabbit, polyclonal) | Proteintech | 25443–1-AP | 1:750 |
| Antibody | EMC4 (rabbit, polyclonal) | Bethyl | A305-752A | 1:500 |
| Antibody | Dengue virus prM protein (rabbit, polyclonal) | Genetex | GTX128093 | 1:1000 |
| Antibody | Dengue virus capsid protein (rabbit, polyclonal) | Genetex | GTX103343 | 1:1000 |

*Continued on next page*

*Continued*

| Reagent type (species) or resource | Designation | Source or reference | Identifiers | Additional information |
|---|---|---|---|---|
| Antibody | Dengue virus NS2B (rabbit, polyclonal) | Genetex | GTX124246 | 1:1000 |
| Antibody | Dengue virus NS3 protein (mouse, monoclonal) | Genetex | GTX629477 | 1:1000 |
| Antibody | Dengue virus NS4B protein (rabbit, polyclonal) | Genetex | GTX124250 | 1:1000 |
| Antibody | Dengue virus Type 2 NS5 protein (mouse, monoclonal) | Genetex | GTX629447 | 1:1000 |
| Antibody | GFP (rabbit, polyclonal) | Genetex | GTX113617 | 1:5000 |
| Antibody | FLAG (mouse, monoclonal) | Sigma | F1804 | 1:2000 |
| Recombinant DNA reagent | EMC2 cDNA | GenScript | clone OHu30604 | |
| Recombinant DNA reagent | EMC4 cDNA | GenScript | clone OHu00964 | |
| Commercial assay or kit | Renilla Luciferase Assay system | Promega | E2810 | |
| Commercial assay or kit | Power SYBR Cells-to-CT kit | ThermoFisher | 4402953 | |
| Chemical compound, drug | MK-0608 (2'-C-Methyl cytidine) | Carbosynth | NM07918 | working concentration: 25 µM |
| Chemical compound, drug | Bortezomib | Selleckchem | S1013 | working concentration: 1–50 nM |
| Chemical compound, drug | Cycloheximide | Sigma | C7698 | working concentration: 100 µg/ml |
| Software, algorithm | Prism 8 | GraphPad | | |
| Software, algorithm | FlowJo | FlowJo | | |

## Cell culture and viruses

Huh7.5.1 (obtained from Peter Sarnow, original gift from Frank Chisari) and HEK293FT (Thermo Scientific) were cultured in DMEM (Gibco) supplemented with 10% fetal bovine serum (Omega Scientific), penicillin/streptomycin (Gibco), non-essential amino acids (Gibco) and L-glutamine (Gibco) at 37C and 5% $CO_2$. Parental cell lines were tested negative for mycoplasma and were authenticated by STR profiling. The following virus strains were used: DENV serotype 2 strain 16681 (stocks kindly provided by Caleb Marceau), DENV-Luc (stocks kindly provided by Jan Carette) (*Marceau et al., 2016*), WNV Kunjin strain CH16532 (stocks kindly provided by Jan Carette), ZIKV PRVABC59 (stocks

kindly provided by Jan Carette) and HCV JFH1 (stocks kindly provided by Jan Carette). Viral titers were determined by plaque- or focus-forming assay.

## Chemical compounds

MK-0608 (2'-C-Methylcytidine) was purchased from Carbosynth and used at a concentration of 25 µM. Bortezomib was obtained from Selleckchem and used at concentrations 1–50 nM. For all experiments, DMSO controls were included. Cycloheximide was purchased from Sigma and used at 100 µg/ml for Ribosome Profiling experiments.

## Generation of knockout cell lines

Oligos containing sgRNA sequence were annealed and ligated into pX458 (gift from Feng Zhang; Addgene plasmid # 48138). Cells were transfected with pX458 using Lipofectamine 3000 (Invitrogen) and two days later GFP positive cells were single-cell sorted into 96-well plates using a Sony SH800 cell sorter. For genotyping, genomic DNA was isolated from obtained clones using QuickExtract (Lucigen), the sgRNA-targeted sites PCR amplified and the products Sanger-sequenced. Obtained sequences were compared to reference sequences for indel mutations. In the case of heterozygous mutations, TIDE was used to deconvolute sequencing traces (*Brinkman et al., 2014*). A list of all used sgRNA oligo and genotyping primer sequences can be found in *Supplementary file 1*.

## Dengue Luciferase assay

Huh7.5.1 or HEK293FT cells were seeded in 96-well plates and infected with DENV-Luc at an moi of 0.01. Cells were harvested in Renilla Lysis buffer and luminescence was measured using Renilla Luciferase Assay system (Promega) on a Spectramax i3x Multi-Mode Microplate Reader (Molecular Devices).

## Quantitative RT-PCR

Huh7.5.1 or HEK293FT cells were plated in 96-well plates (in triplicates for each condition) and infected with an moi of 0.5 for all viruses. Cells were then harvested at 30hpi for ZIKV, 48hpi for DENV and WNV, and 72hpi for HCV using the Power SYBR Cells-to-CT kit (Invitrogen). After reverse transcription, quantitative PCR was performed on a Bio-Rad CFX96 Touch system. All viral RNA levels were normalized to 18S levels. For viral internalization assay, 96-well plates containing WT or EMC4 KO Huh7.5.1 cells were put on ice before infection with DENV at an moi of 30. Cells were incubated for 1 hr before moving them to a 37C incubator. At each harvest timepoint, cells were washed three times with PBS and lysed as described above. All qPCR primer sequences can be found in *Supplementary file 1*.

## Lentiviral complementation

EMC2 (GenScript; clone OHu30604) and EMC4 cDNA (GenScript; clone OHu00964) were PCR amplified using tgtggtggaattctgcagataccATGGCGAAGGTCTCAGA/CGGCCGCCACTGTGCTGGA TTTACTTATCGTCGTCATCCTTGTAATCAGACTGGGTGATC and tgtggtggaattctgcagataccA TGACGGCCCAGGGG/CGGCCGCCACTGTGCTGGATTTACTTATCGTCGTCATCCTTGTAA TCCAAAAGCAGTCCT, respectively, and cloned into EcoRV-cut pLenti CMV Puro DEST (w118-1) (gift from Eric Campeau and Paul Kaufman; Addgene plasmid # 17452) using NEBuilder HiFi DNA Assembly Master Mix (New England BioLabs). Lentivirus was produced in HEK293FT, respective KO cells transduced with filtered, lentiviral containing supernatant and selected using 2–4 µg/ml puromycin (Gibco) for 3–4 days.

## Replicon assay

DENV replicon plasmid (*Marceau et al., 2016*) was linearized using XbaI restriction enzyme. Replicon RNA was generated using the MEGAscript T7 Kit (Invitrogen) with the reaction containing 5mM m7G(5')ppp(5')G RNA Cap Structure Analog (New England BioLabs). Resulting RNA was purified by lithium chloride precipitation. For electroporation, 2 million WT or EMC4 KO HEK293FT cells were washed twice in PBS, resuspended in 100 µl SF Nucleofector solution (Lonza), mixed with 4 µg replicon RNA, transferred to a 100 ul nucleocuvette and pulsed using program CM-130 on an Amaxa 4D-Nucleofector X Unit (Lonza). Cells were then resuspended in antibiotic-free medium, distributed into

96-wells and lysed at different timepoints post-electroporation using Renilla Lysis buffer. Luminescence was measured using Renilla Luciferase Assay system (Promega) on a Spectramax i3x Multi-Mode Microplate Reader (Molecular Devices).

## Immunoblotting

Cell were lysed using Laemmli SDS sample buffer containing 5% beta-mercaptoethanol and boiled for 10 min. Lysates were separated by SDS-PAGE on pre-cast Bio-Rad 4–15% poly-acrylamide gels in Bio-Rad Mini-Protean electrophoresis system. Proteins were transferred onto PVDF membranes using Bio-Rad Trans-Blot Turbo transfer system. PVDF membranes were blocked with PBS buffer containing 0.1% Tween-20% and 5% non-fat milk. Blocked membranes were incubated with primary antibody diluted in blocking buffer and incubated overnight at 4C on a shaker. Primary antibodies were detected by incubating membranes with 1:10000 dilution of HRP-conjugated (Southern Biotech) or IRDye-conjugated (LI-COR) secondary anti-mouse and anti-rabbit antibodies for 1 hr at room temperature. Blots were visualized using a ChemiDoc MP Imaging System (Bio-Rad). The following primary antibodies (and their dilutions) were used in this study: p84 (Genetex, GTX70220) at 1:2000, GAPDH (SCBT, sc-32233) at 1:2000, EMC2/TTC35 (Proteintech, 25443–1-AP) at 1:750, EMC4 (Bethyl, A305-752A) at 1:500, Dengue virus prM protein (Genetex, GTX128093) at 1:1000, Dengue virus Capsid protein (Genetex, GTX103343) at 1:1000, Dengue virus NS2B (Genetex, GTX124246) at 1:1000, Dengue virus NS3 protein (Genetex, GTX629477) at 1:1000, Dengue virus NS4B protein (Genetex, GTX124250) at 1:1000, Dengue virus Type 2 NS5 protein (Genetex, GTX629447), GFP (Genetex, GTX113617) at 1:5000 and FLAG M2 (Sigma, F1804) at 1:2000.

## Ribosome profiling

WT and EMC4 KO HEK293FT were plated in 10 cm dishes and infected with DENV at an of 100 for the 18hpi harvest and an moi of 0.5 for the 44hpi harvest. Cells were treated with 100 µg/ml cycloheximide for 2 min, washed once with polysome gradient buffer (20 mM Tris pH 7.5, 150 mM NaCl, 5 mM $MgCl_2$, 100 µg/mL cycloheximide, 1 mM DTT) and then lysed using 500 µl of polysome lysis buffer (20 mM Tris pH 7.5, 150 mM NaCl, 5 mM $MgCl_2$, 1% Triton X-100, 1 mM DTT, 100 µg/mL cycloheximide, 24 U/ml Turbo DNase (Ambion)) per plate. Cells/lysates were collected using cell scrapers and incubated on ice for 15 min. Crude lysates were centrifuged at 20,000 x g for 2 min at 4C and the cleared polysome-containing lysates flash frozen by immersion in liquid nitrogen and stored at −80C until further processing. For ribosomal footprinting, 30 µg total RNA diluted in polysome buffer to a total volume of 200 µl were treated with 750 U/ml RNaseI (Ambion) for 45 min at 25C with shaking (300 rpm). RNase activity was quenched by adding 10 ul SuperaseIN (Invitrogen) and placing samples on ice. Note that 10% of the lysates were saved for RNA-seq library preparation (see below). A 1 ml sucrose cushion was added to the bottom of a 13 mm x 56 mm thick-wall poly-carbonate tube (Beckman Coulter) for each sample and the footprint digests were layered on top. Tubes were ultracentrifuged at 100,000 rpm for 4 hr at 4C in a TLA100.3 rotor (Beckman Coulter). After spin, supernatant was removed and the monosome-containing pellet was resuspended in 300 µl Trizol LS (Life Technologies) for RNA extraction using the Direct-Zol kit (Zymo). Ribosome Profiling NGS libraries were generated as previously described (*Shurtleff et al., 2018*).

For corresponding RNA-seq libraries RNA was extracted from the remaining undigested cleared lysates using Trizol and the Direct-Zol kit. 1 µg purified total RNA was used as input for the Truseq stranded total RNA Gold library prep (Illumina). Ribosomal RNA was depleted using Ribo-Zero H/M/R Gold and depleted RNA fragmented for 8 min at 94C. Following first and second strand synthesis products were purified using SPRI beads. KAPA dual-indexed adapters (Kapa Biosystems, KK8722) were used for adapter ligation. Finally, PCR enriched products were cleaned up using SPRI. Ribosome Profiling and RNA-seq libraries were analyzed on Bioanalyzer or TapeStation (Agilent) before sequencing on a NextSeq 500 platform (Illumina) in separate runs with 400M reads output.

## Analysis of Ribosome Profiling data

Reads from Ribosome Profiling or RNA-seq were preprocessed by removing low-quality reads and trimming of adaptor sequences. Next, ribosomal RNA reads were removed using Bowtie. Remaining reads were aligned to the human reference genome file also containing the genomic DENV2 16681 sequence using Tophat to generate read count tables with RPKMs for each transcript. A cutoff of at

least 5 RPKMs was used for further analysis of transcripts/footprint counts, and $\log_2$ fold-changes (FC) were calculated between corresponding EMC4 KO and WT samples for both Ribosome Profiling and RNA-seq. To calculate translation efficiency (TE), the Ribosome Profiling RPKMs were normalized by the RNA-seq RPKMs for each transcript.

To analyze the footprint distribution of Ribosome Profiling reads across the DENV genome, DENV specific reads were mapped to the DENV2 16681 reference sequence using Geneious Prime and coverage at each nucleotide position was determined.

## Dual fluorescence cytometry assay

eGFP was amplified from pX458 with adding overhang to plenti-CMV-Puro (EcoRV cut site), a Kozak sequence and ATG start codon for the G2C-only construct or with an additional overhang to GlySer-linker sequence for fusion with DENV fragments. T2A-mCherry was amplified from pX458 where eGFP had been replaced with mCherry. Different regions of the DENV genome were amplified from the DENV2 16681 infectious clone plasmid (gift from Karla Kirkegaard) with addition of overhang sequences to plenti-CMV backbone, Kozak sequence and ATG start codon on 5'end and to GlySer-linker-GFP fragment on 3'end. ADRB1 was amplified from pcDNA3 Flag beta-1-adrenergic-receptor (gift from Robert Lefkowitz; Addgene plasmid # 14698). All fragments contained ~ 20 bp overhangs and were assembled into EcoRV cut pLenti CMV Puro DEST (w118-1) using NEBuilder HiFi DNA Assembly Master Mix (New England BioLabs) or Gibson-like master mix (Berkeley MacroLabs) to generate G2C reporter expression constructs. All constructs were verified by Sanger sequencing. Primer sequences can be found in *Supplementary file 1*.

For the G2C dual fluorescence assay, cells were seeded in 24- or 12-well plates and transfected with G2C constructs using Lipofectamine 3000 (Invitrogen). 24–48 hr later cells were trypsinized, filtered (70 μm), and analyzed by flow cytometry using a Cytoflex S flow cytometer (Beckman Coulter). At least 5000 and 10,000 mCherry positive cells were recorded for Huh7.5.1 and HEK293FT, respectively. For analysis, cells were gated based on FSC/SSC, FSC-H/FSC-A (singlets) and finally ECD (mCherry) and FITC (eGFP) using FlowJo 10. Raw fluorescence data for all gated fluorescent cells were exported and the eGFP:mCherry intensity ratios were calculated for individual cells. All fluorescence ratios were normalized to the mean of the G2C-only construct. Distributions of fluorescence ratios were plotted as histograms using GraphPad Prism 8.

## Viral adaptation

WT DENV serotype 2 strain 16681 was serially passaged every 3–4 days onto newly plated EMC4 KO Huh7.5.1 cells by transferring increasingly diluted supernatant from infected cells (starting at 2-fold dilutions for the first three passages, then stepwise increase of dilutions to 100-fold for the final passages). Six independent passaging experiments were started and cytopathic effects were observed in three of the replicates after approximately 20–22 passages. Supernatant was then harvested, filtered through a 0.45 um filter to remove cellular debris, and RNA was extracted using Trizol LS and Direct-Zol kit (Zymo). Additionally, RNA was extracted from WT DENV stock. To prepare sequencing libraries, Truseq stranded total RNA Gold library kit (Illumina) was used. Isolated RNA was fragmented for 4 min at 94C without prior Ribo-Zero depletion, followed by first and second strand synthesis, and adapter ligation. Fragments were amplified in presence of SYBR Green on Bio-Rad CFX96 and reaction was stopped for each sample before it reached plateau (~20–24 cycles). Libraries from different virus isolates were pooled and sequenced on iSeq 100 (Illumina) with 2 × 75 bp reads.

For analysis, Geneious Prime was used: Fastq files were imported as paired reads and Truseq adapters and low-quality sequences (minimum 20) were trimmed using BBDuk. Next, reads were mapped to the WT 16681 reference genome using the Geneious Mapper at medium-low sensitivity and with five iterations for fine-tuning. All samples had a minimum coverage of 1000 at all nucleotide positions. Finally, SNPs between adapted and WT stock viruses were called with a minimum variance frequency of 80%. To compare acquired non-synonymous point mutations to a large dataset of sequenced DENV2 strains, 1359 complete genome sequences from NCBI were mapped to the DENV2 16681 strain containing adapted mutations.

## Generation of EMC adapted DENV-Luciferase

DENV-Luc infectious clone containing adapted mutations were generated by PCR amplification from WT DENV-Luc infectious clone plasmid (gift from Jan Carette). For DENV-Luc(A6665G) fragments were amplified using GAAGAGTGATGGTTATGGTAGGC/CTGTATTTGTGCGCACCATAGGAGGATG and CATCCTCCTATGGTGCGCACAAATACAG/GTGATCTTCATTTAAGAATCCTAGGGCTTC. For DENV-Luc(A7558T) fragments were amplified using GAAGAGTGATGGTTATGGTAGGC/CCCCTTCTTGTGTAGGTTGTGTTCTTC and GAAGAACACAACCTACACAAGAAGGGG/GTGATCTTCATTTAAGAATCCTAGGGCTTC.

For DENV-Luc(A6665G, A7558T) fragments were amplified using GAAGAGTGATGGTTATGGTAGGC/CTGTATTTGTGCGCACCATAGGAGGATG, CATCCTCCTATGGTGCGCACAAATACAG/CCCCTTCTTGTGTAGGTTGTGTTCTTC, and GAAGAACACAACCTACACAAGAAGGGG/GTGATCTTCATTTAAGAATCCTAGGGCTTC. PCR products were cloned into NarI and AvrII cut WT DENV-Luc infectious clone using NEBuilder HiFi DNA Assembly Master Mix (New England BioLabs) and transformed into Stbl3 bacteria (Berkeley MacroLabs). Constructs were verified by Sanger sequencing. Plasmids were linearized with XbaI and RNA was generated by in-vitro transcription using the MEGAscript T7 Kit (Invitrogen) with the reaction containing 5mM m7G(5')ppp(5')G RNA Cap Structure Analog (New England BioLabs). Resulting RNA was purified by lithium chloride precipitation and transfected into BHK-21 cells using Lipofectamine 3000 for viral production. Supernatants were harvested 6–9 days post-transfection.

## Focus-forming unit assay

Huh7.5.1 cells were seeded in 24-well plates. Next day, cells were infected with DENV stocks at serial dilutions from $10^{-1}$ to $10^{-6}$ in a total volume of 500 µl. 30 hr later, cells were fixed with 4% PFA in PBS. Cells were washed three times with PBS and then blocked for 15 min at room temperature in blocking buffer (PBS containing 1% saponin, 1% Triton X-100, 5% fetal bovine serum, and 0.1% azide). Fixed cells were incubated with mouse 4G2 antibody (Novusbio, # D1-4G2-4-15) diluted 1:1000 in blocking buffer for 1 hr with gentle shaking. Primary antibody was washed off with three PBS washes and then cells were incubated with Alexa-Fluor 488 anti-mouse antibody (Life Technologies) diluted 1:1000 for 30 min. Secondary antibody was washed off with two PBS washes. Lastly, cells were stained with Hoechst diluted 1:2000 in PBS and fluorescent colonies were counted under the microscope.

## Co-immunoprecipitation

HEK293FT EMC4 KO cells complemented with FLAG-tagged EMC4 were seeded in a 10 cm dish and infected with DENV (moi = 1) for 96 hr. Cells were washed with PBS and then collected in 1 ml Pierce RIPA buffer (Thermo Scientific) containing 1 mM PMSF (Thermo Scientific), 5 mM EDTA and 1x Halt Protease Inhibitor Cocktail (Thermo Scientific) with help of cell scrapers. Cells were lysed on ice for 1 hr with occasional vortexing and subsequently centrifuged at 15,000 x g for 10 min at 4C to clear lysates. For co-immunoprecipitation, clarified lysates were incubated at 4C overnight with Anti-FLAG M2 Magnetic Beads (Sigma) while rotating. Beads were washed three times using TBS-T and eluted by boiling in 2x Laemmli buffer. Eluates were loaded for SDS-PAGE and stained for viral proteins.

## Protease protection assay

NS4A-4B was amplified starting from NS4A and ending as indicated in *Figure 6D* with addition of an upstream overhang to plenti-CMV-Puro, a Kozak sequence and ATG start codon, and a downstream overhang to eGFP. Fragments were assembled into EcoRV-cut plenti-CMV-Puro using Gibson-like master mix (Berkeley MacroLabs). Plasmids were then transfected into WT and EMC4 KO cells, trypsinized after 24 hr and pelleted by centrifugation at 400xg for 5 min. Pellets were resuspended in PBS, split into two tubes and pelleted again. Pellets were either resuspended in KHM buffer (110 mM potassium acetate, 20 mM HEPES, 2 mM MgCl$_2$) or KHM buffer containing 50 µM digitonin (Sigma) and 50 µg/ml proteinase K (Qiagen) and incubated on ice for 45 min. Subsequently, PMSF (Sigma) was added to a final concentration of 10 mM. After 10 min, 2x Laemmli buffer containing 5% beta-mercaptoethanol was added and samples were boiled for 10 min. Lysates were analyzed by immunoblotting as described above using the anti-GFP antibody.

## Deglycosylation assay

An opsin tag with the amino acid sequence GPNFYVPFS<u>NKT</u>G (containing a sequon for N-glycosylation) (*Buentzel and Thoms, 2017*) was introduced downstream of NS4A TMD2 at amino acid position 99 of NS4A. Constructs contained either the WT sequence or the A6665G mutation and were FLAG-tagged at the N-terminus of NS4A. WT and EMC4 KO cells were transfected, harvested in Pierce RIPA buffer (Thermo Scientific) containing 1x Halt Protease Inhibitor Cocktail (Thermo Scientific) after 24 hr, and deglycosylation assay was performed using PNGase F (New England BioLabs) according to the manufacturer's instructions. Briefly, lysates were mixed with Glycoprotein Denaturing Buffer and incubated at 98C for 10 min. Subsequently, 10% NP-40 and GlycoBuffer two were added, samples split in halve and incubated at 37C for 1.5 hr with or without PNGase F. Finally, 4x Laemmli buffer containing 5% beta-mercaptoethanol was added, samples were boiled for 10 min and analyzed by immunoblotting using anti-FLAG M2 antibody (Sigma).

## Statistical analysis and reproducibility

GraphPad Prism 8 was used to analyze data and perform statistical tests. Unpaired *t*-tests were applied to measure statistical significance of differences in DENV infection between WT and EMC KO cells. Infection experiments were performed with three biological replicates each. Biological replicates are defined as independent infections and measurements of separate wells. Each set of infection experiments containing three biological replicates was repeated twice and one representative is shown in the figures. Immunoblot and flow cytometry experiments were performed at least twice and one representative is shown.

## Acknowledgements

We like to thank the following people: Jan Carette for discussions, providing reagents and critically reading the manuscript; Caleb Marceau and Kelsey Hickey for technical advice and providing reagents; members of the Biohub Infectious Disease Initiative and the Weissman lab for discussions and technical advice; the Biohub Genomics Platform for help with sequencing. ASP was supported by the Chan Zuckerberg Biohub. MJS is a Howard Hughes Medical Institute Fellow of the Helen Hay Whitney Foundation and JSW is an Investigator of the Howard Hughes Medical Institute.

## Additional information

### Funding

| Funder | Author |
| --- | --- |
| Howard Hughes Medical Institute | Jonathan S Weissman |
| Helen Hay Whitney Foundation | Matthew J Shurtleff |

The funders had no role in study design, data collection and interpretation, or the decision to submit the work for publication.

### Author contributions

Ashley M Ngo, Katerina D Popova, Formal analysis, Validation, Investigation, Methodology, Writing—review and editing; Matthew J Shurtleff, Formal analysis, Funding acquisition, Validation, Investigation, Methodology, Writing—review and editing; Jessie Kulsuptrakul, Validation, Investigation; Jonathan S Weissman, Conceptualization, Resources, Supervision, Funding acquisition, Writing—review and editing; Andreas S Puschnik, Conceptualization, Resources, Formal analysis, Supervision, Funding acquisition, Validation, Investigation, Visualization, Methodology, Writing—original draft, Writing—review and editing

## Author ORCIDs

Matthew J Shurtleff (iD) http://orcid.org/0000-0001-9846-3051
Jonathan S Weissman (iD) http://orcid.org/0000-0003-2445-670X
Andreas S Puschnik (iD) https://orcid.org/0000-0002-9605-9458

## Decision letter and Author response

Decision letter https://doi.org/10.7554/eLife.48469.021
Author response https://doi.org/10.7554/eLife.48469.022

## Additional files

### Supplementary files

• Supplementary file 1. Table of oligonucleotides used in this study: sgRNA oligos, genotyping primers, qPCR primers, primers to generate G2C constructs.
DOI: https://doi.org/10.7554/eLife.48469.016

• Transparent reporting form
DOI: https://doi.org/10.7554/eLife.48469.017

### Data availability

Ribosome profiling/RNAseq data have been deposited as NCBI BioProject under the accession code PRJNA526529.

The following dataset was generated:

| Author(s) | Year | Dataset title | Dataset URL | Database and Identifier |
|---|---|---|---|---|
| Ngo AM, Shurtleff MJ, Popova KD, Kulsuptrakul J, Weissman JS, Puschnik AS | 2019 | Ribosome profiling and RNA-seq of dengue virus infected HEK293 cell lines | https://www.ncbi.nlm.nih.gov/bioproject/?term=PRJNA526529 | NCBI BioProject, PRJNA526529 |

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
