## [Decision Letter]

Thank you for submitting your article "The ER membrane protein complex is required to ensure correct topology and stable expression of flavivirus polyproteins" for consideration by *eLife*. Your article has been reviewed by three peer reviewers, one of whom is a member of our Board of Reviewing Editors, and the evaluation has been overseen by Karla Kirkegaard as the Senior Editor. The following individual involved in review of your submission has agreed to reveal their identity: Sumana Sanyal (Reviewer #2).

Summary:

This is an interesting investigation into the mechanisms for the requirement of EMC for flavivirus replication. EMC was found to be required for proper folding and thus the stability of DENV proteins. The NS4A-NS4B domain was shown to be critical to mediate the interactions of the viral protein with EMC. This manuscript provides a model of EMC-dependent viral protein biogenesis, corroborates independent reports on the function of EMC in flavivirus replication and expands our understanding of the EMC. The very nice Discussion section does a good job of setting the findings within a broader context and put forward a model whereby EMC may directly facilitate biogenesis of a wider variety of transmembrane (TM) proteins than was previously reported, specifically proteins with internal TM domains distant from the protein N and C termini. While all the reviewers found the paper interesting, some revisions, one of them experimental, are needed before it can be published.

Essential revisions:

The new topology assay is great. Analyzing the topology of the adapted NS4A-NS4B in EMC4 KO cells is also needed to support or extend the mechanistic analysis. Does the adaptation mutation now allow correct folding? Or does it allow the virus to grow with incorrect folding? These experiments are necessary to understand the nature of the growth defect and the structure-function relationship of the adaptation mutation. Either answer would be interesting, and necessary to ask whether the complex topology itself is needed or the cytoplasmic presentation of particular sequences.

Minor points:

1) The authors should compare the data in this manuscript and the ones published previously by Lin et al. (Cell Reports, 2019).

2) In 293FT cells, the phenotypes of EMC4 and EMC3 KO were so different (Figure 1B). Does this suggest that EMC4 could act in a manner independent of EMC or EMC could work independently of EMC4? All the later experiments were done with the EMC4 KO cells.

3) In Figure 2A, it appears that it is more the binding than the fusion and release of viral RNA from the endosomes that is being measured. In which case, RT PCR will not provide much information.

4) In Figure 6A, stability of NS4A-4B appears to be the same even in non BZ treated samples. However, in Fig 4B, stability of NS4A-4B is lower in EMC4 KO cell. Please address this discrepancy.

5) The figure legends would benefit from more experimental details.

---

## [Author Response]

Essential revisions:The new topology assay is great. Analyzing the topology of the adapted NS4A-NS4B in EMC4 KO cells is also needed to support or extend the mechanistic analysis. Does the adaptation mutation now allow correct folding? Or does it allow the virus to grow with incorrect folding? These experiments are necessary to understand the nature of the growth defect and the structure-function relationship of the adaptation mutation. Either answer would be interesting, and necessary to ask whether the complex topology itself is needed or the cytoplasmic presentation of particular sequences.

Since a topological defect was observed early in NS4A-4B and membrane proteins are co-translationally translocated across and inserted into the membrane, we initially focused on the A6665G mutation, which is located in TMD2 of NS4A, rather than the A7558T mutation near the C-terminus of NS4B.

First, we defined the topology of wild-type NS4A using a deglycosylation assay to directly compare to a published study, which had previously analyzed the NS4A topology (PMID: 17276984). To probe for stable luminal localization of NS4A TMD2, we introduced an opsin tag containing a N-glycan acceptor site downstream of NS4A TMD2. Luminal localization of the sequon will result in N-glycosylation and thus a band shift as visualized by Western blot after treatment with PNGase F, a deglycosylation enzyme (PMID: 28409466). We observed efficient glycosylation in WT cells (see new Figure 6E) as expected from the previous report. By contrast, almost no glycosylation was detected in EMC4 KO cells indicating an aberrant topology of NS4A. To test whether the adaptive mutations restore the topology or allow the virus to replicate with a different topology in the absence of the EMC, we performed the deglycosylation assay using the NS4A-TMD2-opsin construct containing the A6665G adaptive mutation. As the A6665G mutation led to a significant increase in replication in EMC deficient cells, we would expect any rescue of correct topology to be visible. However, no increase of glycosylation was observed in EMC4 KO cells using the mutant construct (see new Figure 6E). Therefore, we conclude that the adaptive mutation in NS4A does not immediately restore the correct topology after TMD2 but instead allows the virus to replicate with NS4A-4B assuming an aberrant topology.

To expand our results, we additionally attempted to probe the topology using full-length NS4A-4B constructs containing either WT sequence or both adaptive mutations (A6665G in NS4A and A7558T in NS4B). However, our efforts were unsuccessful as introduction of glycan acceptor sites either impaired expression of the constructs or we did not observe efficient glycosylation of luminal sites even in WT cells, most likely due to poor accessibility of certain regions by the oligosaccharyltransferase. This was the case despite introducing sequons in different luminal positions and adding various flexible linkers surrounding the acceptor site to increase accessibility. Moreover, we attempted a different strategy using cysteine scanning, which relies on engineering individual cysteines into cytoplasmic loops and using their chemical reactivity for tagging. Similarly, we did not observe reliable expression of these constructs.

Nevertheless, we think the results regarding the A6665G mutation inside NS4A-TMD2 provide a convincing argument that the wild-type topology is not restored in EMC KO cells. Since membrane proteins are co-translationally translocated across and inserted into the membrane, it is unlikely that the A7558T near the C-terminus of NS4B affects the topology of the upstream TMDs in NS4A and NS4B. However, we cannot fully exclude that the adaptive mutations only restore the wild-type topology in tandem or in the context of full-length viral polyprotein, for example through interactions with other viral proteins as the two mutations were selected by passaging replicating virus.

We have included the new results in Figure 6E and described them in the last paragraph of the Results. We additionally discuss the new findings in the fourth paragraph of the Discussion.

Minor points:

1) The authors should compare the data in this manuscript and the ones published previously by Lin et al. (Cell Reports, 2019).

We thank the reviewers for this suggestion. We have now included a paragraph in the discussion, where we compare our results with the study published by Lin et al (see lines 310-336):

“Recently, another study found that biogenesis of DENV and ZIKV NS4A and NS4B is dependent on the EMC (Lin et al., 2019). While Lin et al. used ectopic expression of individual viral proteins to identify NS4A and NS4B as EMC client proteins, we performed a fluorescent based stability assay expressing different fragments of the viral polyprotein leading to similar conclusions. However, we only observed destabilization when NS4B was preceded by NS4A arguing that the NS4A-4B intermediate triggers an EMC-dependent process. Both studies showed that the EMC physically interacts with NS4B to facilitate stable expression and that, in absence of EMC, NS4B is misfolded and targeted for degradation as demonstrated by proteasomal inhibition. Lin et al. showed that NS4B is still membrane associated in EMC deficient cells, while we provided further molecular insights that NS4A-4B assumes an aberrant topology at the ER membrane. Furthermore, bioinformatic prediction of NS4B transmembrane regions revealed that the two N-terminal domains are only weakly hydrophobic, which could explain the dependence of NS4B on the EMC for stable expression (Lin et al., 2019). This is consistent with the systematic analysis of cellular EMC client proteins, which showed an underrepresentation of hydrophobic and an overrepresentation of charged and aromatic amino acids within their TMDs compared to all TMDs in the proteome (Shurtleff et al., 2018). Consequently, Lin et al. demonstrated that substitution of all polar and charged residues inside TMD1 and TMD2 of NS4B to hydrophobic leucines led to complete rescue of ectopic NS4B expression in EMC depleted cells. However, it remains unclear whether these substitutions would allow for viral replication in the full polyprotein context. By contrast, our EMC KO adapted mutations do not reside in TMD1 and TMD2 of NS4B but in NS4A TMD2 and near the C-terminus of NS4B, and were shown to increase NS4A-4B expression levels as well as restore viral replication. Overall, both studies are complementary and in large agreement, and together provide a detailed picture of how the EMC ensures stable expression of two viral multi-pass transmembrane proteins.”

2) In 293FT cells, the phenotypes of EMC4 and EMC3 KO were so different (Figure 1B). Does this suggest that EMC4 could act in a manner independent of EMC or EMC could work independently of EMC4? All the later experiments were done with the EMC4 KO cells.

The ER membrane protein complex (EMC) consists of 10 subunits (EMC1-10) in mammalian cells. Independently conducted genetic screens for flavivirus host factors in different cell types, such as Huh7.5.1 (PMID: 27383987), 293T (PMID: 27383988), HAP1 (PMID: 31384002), HeLa H1 (PMID: 27342126) or neural progenitor cells (PMID: 31019072), all identified multiple subunits of the EMC including EMC3 and EMC4. This suggests that the entire complex is important for viral replication and that knockout of one subunit is generally sufficient to disrupt EMC function.

Quantitative assays for viral replication in Figure 1 revealed differences in the strength of the phenotypes of different EMC KO mutations in different cell types. While knockout of any subunit (EMC1-6) in Huh7.5.1 cells led to a >1,000x reduction of dengue replication (Figure 1A), we observed differences in the degree of viral inhibition in EMC KO in 293FT cells (Figure 1B). As stated in the text, we speculate that this could be due to the degree of disruption of the entire protein complex. Knockout of EMC3 in 293FT may result in a partially intact complex while deletion of the EMC4 subunit completely destroys EMC activity. We highlight this as follows: “The differences in the extent of the phenotypes may be attributed to the presence of compensatory pathways or the degree of disruption of the entire protein complex between the two cell types upon knockout of a single subunit.”

The majority of experiments were performed with EMC4 KO cells to have the largest dynamic range for phenotypic readouts in both Huh7.5.1 and HEK293FT cells.

3) In Figure 2A, it appears that it is more the binding than the fusion and release of viral RNA from the endosomes that is being measured. In which case, RT PCR will not provide much information.

We agree with the reviewers that RT-qPCR of viral RNA upon incubation of virus with cells at 4C followed by incubation at 37C cannot distinguish between virions attached to the cell surface and virions that are internalized and released from the endosomes. We have changed the wording in the main text accordingly.

Nevertheless, the viral binding/uptake data together with the replicon results still lead to the conclusion that the EMC is required for intracellular steps of the viral life cycle. In the replicon assay (Figure 2B), in vitro transcribed viral RNA is electroporated into the cells thus bypassing the requirement for viral fusion and release into the cytosol. Thus, differences in luminescence levels starting at 12h post-electroporation between WT and EMC KO cells indicate that viral replication and/or translation are impaired. However, we cannot completely rule out an additional effect on virus infection post-attachment (e.g. fusion or release from endosomes).

4) In Figure 6A, stability of NS4A-4B appears to be the same even in non BZ treated samples. However, in Fig 4B, stability of NS4A-4B is lower in EMC4 KO cell. Please address this discrepancy.

We thank the reviewers for pointing out this discrepancy. In the G2C reporter assay (Figure 4B) we used a NS4A-4B construct with a C-terminal fusion to GFP-2A-mCherry and we calculated the ratio of the GFP-fusion protein to mCherry as a measure of stability. It has been shown that cleavage between NS4A and NS4B can occur in absence of the viral protease and is mediated by the signal peptidase at the 2K-NS4B junction (PMID: 1531368). Therefore, we think that the destabilized form that is measured in the G2C assay is likely to be the cleaved NS4B-eGFP fusion protein and not NS4A-NS4B-eGFP, which would be in agreement with the Western blot data in Figure 6A. This is also consistent with the immunoblot in Figure 6D, where after transfection of NS4A-4B-eGFP we detected NS4B-eGFP at the molecular weight expected for NS4B+eGFP, thus after cleavage from NS4A.

5) The figure legends would benefit from more experimental details.

We have now added more experimental details to the figure legends in the revised manuscript.